# Salt and Drought Stress Responses in Cultivated Beets (*Beta vulgaris* L.) and Wild Beet (*Beta maritima* L.)

**DOI:** 10.3390/plants10091843

**Published:** 2021-09-05

**Authors:** Seher Yolcu, Hemasundar Alavilli, Pushpalatha Ganesh, Madhusmita Panigrahy, Kihwan Song

**Affiliations:** 1Faculty of Engineering and Natural Sciences, Sabanci University, Istanbul 34956, Turkey; 2Department of Bioresources Engineering, Sejong University, Seoul 05006, Korea; 3Department of Plant Biotechnology, M. S. Swaminathan School of Agriculture, Centurion University of Technology and Management, Khurda 761211, Odisha, India; pushpabhagyalakshmi@gmail.com; 4Biofuel & Bioprocessing Research Center, Institute of Technical Education & Research, Siksha ‘O’ Anusandhan Deemed to Be University, Bhubaneswar 751030, Odisha, India; mpanigrahy@niser.ac.in

**Keywords:** *Beta vulgaris* L., *Beta maritima* L., drought, salinity, stress tolerance, stress evaluation parameters, stress-responsive genes, noncoding RNAs

## Abstract

Cultivated beets, including leaf beets, garden beets, fodder beets, and sugar beets, which belong to the species *Beta vulgaris* L., are economically important edible crops that have been originated from a halophytic wild ancestor, *Beta maritima* L. (sea beet or wild beet). Salt and drought are major abiotic stresses, which limit crop growth and production and have been most studied in beets compared to other environmental stresses. Characteristically, beets are salt- and drought-tolerant crops; however, prolonged and persistent exposure to salt and drought stress results in a significant drop in beet productivity and yield. Hence, to harness the best benefits of beet cultivation, knowledge of stress-coping strategies, and stress-tolerant beet varieties, are prerequisites. In the current review, we have summarized morpho-physiological, biochemical, and molecular responses of sugar beet, fodder beet, red beet, chard (*B. vulgaris* L.), and their ancestor, wild beet (*B. maritima* L.) under salt and drought stresses. We have also described the beet genes and noncoding RNAs previously reported for their roles in salt and drought response/tolerance. The plant biologists and breeders can potentiate the utilization of these resources as prospective targets for developing crops with abiotic stress tolerance.

## 1. Introduction

According to the new system of classification for cultivated plants, cultivated beets are divided into four cultivar groups, including leaf beets (e.g., Swiss chard), garden beets (red beets), fodder beets (forage beets), and sugar beets [1]. These economically important edible beets belong to the *Amaranthaceae* family, and *Beta vulgaris* L. ssp. *vulgaris* sub-species [1,2]. All beets have originated from a common progenitor, *Beta vulgaris* L. ssp. *maritima*, also known as *Beta maritima* L. (sea beet or wild beet) [3,4]. Among the beets, garden beets and Swiss chard (*B. vulgaris* L. ssp. *vulgaris* var. *cicla*) are consumed as vegetables and fodder beets as fodder for the livestock [1]. The red juice from Swiss chard and red beet is an important source of natural pigments (e.g., betalains), which play a role in free-radical scavenging and have economic value due to their use in the health, pharmaceutical, and food industries [5,6]. Sugar beet stands as the second largest source of refined table sugar after sugar cane, which accounts for approximately 30–40% of global sugar production [7,8,9]. In addition to its being the strength of the sugar industry, the processed wastes and other by-products are used for the production of food additives, bioethanol, biodegradable polymers, and biofertilizers [10,11].

Salinity and drought are among the major abiotic stresses that limit crop growth and production [12]. Soil salinization is increasing due to climate change, sea level rise, irrigation with saline water, and soil erosion across the globe [13], and influences beet productivity negatively [14]. Drought is another major constraint for beet cultivation in the temperate climatic zones, where beet cultivation mainly depends on seasonal rainfall [15]. Although beets are highly salt- and drought-tolerant crops [16,17,18], both stress factors impinge yield loss in beet production and cause growth retardation [19,20,21,22,23]. Hence, a thorough understanding of the abiotic stress-evading strategies at the morphological, physiological and molecular levels for plants is the need of the hour [12]. The existence of genetic diversity among the different stress-tolerant cultivars is considered as a boon for any crop improvement program [24]. The stress-tolerant beet varieties are considered to have genetic potential to maintain growth in response to stress conditions [25]. However, in the case of the sugar beet cultivar group, the genetic diversity is lower when compared to *B. maritima* [26,27]. With the availability of genome sequences of sugar beet (*B. vulgaris*) and wild beet (*B. vulgaris* ssp. *maritima*), they can be good sources of stress-related studies in crops [28,29]. Moreover, isolation of highly salt- and drought-tolerant beet genotypes [16,30,31] and beets’ capability of growing in reclaimed lands, which are adversely affected by salt, sodicity, and poor nutrient availability [32,33,34], will help plant breeders to develop stress-resistant traits and propagate the genetic variability in beets. By virtue of its salt stress tolerance, *B. vulgaris* is an excellent choice for ongoing cultivation in sub-tropical saline soils [35]. Similarly, intercrossing of stress-resistant sugar beet varieties with disease-resistant ones is also an alluring approach to develop stress tolerance in beets. For example, a monosomic sugar beet addition line M14 with tolerance to several stresses such as salt, drought, and cold was generated by an intercross between *B. vulgaris* and a stress-tolerant wild species *B. corolliflora* Zoss. [36,37]. Thus, M14 line is an important genetic resource for the isolation of beneficial genes from wild plants to study stress tolerance mechanisms as well as to generate stress-tolerant beet genotypes [37]. Additionally, the wild beet (*B. maritima* L.), being spatially distributed across the coasts of Mediterranean Sea and the European North Atlantic Ocean, exhibits higher adaptability to saline conditions than other beet varieties during seed germination and seedling stages [17,31,38]. It is used as a donor in breeding programs to improve stress tolerance in modern beet varieties against several pests and diseases [3]. As the wild beet confers numerous stress tolerant traits, it can serve as a valuable resource for crop improvement under abiotic stress [38,39].

In an effort to understand and evaluate the alterations in beets under abiotic stresses and to identify stress-tolerant cultivars, many studies reported differential responses of beet cultivars to salt and drought at morphological, physiological, biochemical, and molecular levels [40,41,42]. Some of the findings also depicted the effects of expressing stress-responsive beet genes in heterologous genomes [43,44,45,46]. However, a comprehensive review summarizing the responses and tolerance mechanisms of cultivated beets (*B. vulgaris* L.) and their ancestor, wild beet (*B. maritima* L.), to salt and drought is lacking. This review explains the morpho-physiological, biochemical, and molecular mechanisms in beet cultivars and their halophytic relative, wild beet under salt and drought stress.

## 2. Morpho-Physiological, Biochemical, and Molecular Changes under Salinity and Drought

Among beet cultivar groups, sugar beet, fodder beet, and red beet (*B. vulgaris* L.) are known as salt-tolerant crop plants, which can tolerate salinity ranging from ~40 to 120 mM NaCl [16,17,37,47]. In addition to salt tolerance, beets display better tolerance to water deficit compared to other grain crops [18,23,48,49]. Major abiotic stress factors such as salinity and drought in beets generally cause various morpho-physiological alterations such as growth retardation, wilting of leaves, reduction in stomatal conductance, photosynthetic rate and transpiration, decline in relative water content (RWC), leaf photosynthetic pigments, lower root biomass, membrane damage through lipid peroxidation, accumulation of compatible solutes, lower white sugar yield, and enhancement of specific leaf weight and succulence index [18,30,31,50,51,52,53,54,55]. In beets, yield reductions after drought might be due to changes in RWC and water potential in leaves [56], limited leaf growth and CO_2_ assimilation [57]. Stomata closure and the reduction of RWC, which occur during the early stage of drought stress, lead to photosynthetic disruption and yield loss in sugar beet [58]. Growth reduction after salt stress might be related to ion toxicity as the photosynthesis levels remain high even under high salinity [59]. According to Munns’ theory, salt stress initially causes osmotic alterations, which decrease the absorption of water (osmotic phase), and then ionic stress due to the accumulation of salts in leaves [60]. In the following sections, we have summarized salt and drought stress responses at different developmental stages of beet, physiological, biochemical, and molecular changes, and the selection parameters of salt- and drought-tolerant beet cultivars from multiple studies. In addition, we have briefly outlined the differences of stress responses in cultivated beets and wild beet. In Table 1, we summarize the effects of salt and drought stress on cultivated beets and wild beet at morpho-physiological and biochemical levels.

### 2.1. A General Overview of Salt and Drought Stress Responses at Different Developmental Stages of Beets

The initial eight-weeks of growth associated with seedling vigor in sugar beet is considered to be the most crucial stage [79], which determines field emergence and stand establishment. Salt and drought stress negatively affect germination and seedling development in sugar beet [4,80] by affecting ionic balance, resulting in hyperosmotic stress and oxidative damage [25,70,81]. Similarly, a highly significant adverse impact of salt stress was observed on germination percentage and index in fodder beet and red beet as well [82,83]. However, three Portuguese wild beet varieties [(Comporta (CMP), Oeiras (OEI) and Vaiamonte (VMT)] can withstand salinity stress at the germination and seedling stages [17,38]. The CMP was found to be capable of initiating and maintaining radicle emergence, even under high salinity [17]. Under greenhouse conditions, salinity stress resulted in a drop in germination rate, and yield, and enhanced mortality in seedlings of sugar beet genotypes [22]. Mostafavi (2012) found that six sugar beet genotypes showed a decrease in germination percentage as the salt stress was increased. This adverse effect of salinity on beet germination is due to ion toxicity [81]. Contrastingly, seed germination in sugar beet was found to be tolerant to salt (200 mM NaCl) or drought (300 mM mannitol) stress [84,85]. Although sugar beet seeds could germinate at high NaCl concentrations, survival till mature stage was severely affected, implying that sugar beet varieties selected for breeding purposes need to be tolerant to salt stress at each developmental stages [22]. Still, cultivated beets are more salt-tolerant than other crops at the vegetative stage under salinity. For instance, Swiss chard was found to be the most salt-tolerant vegetable among different leafy vegetables, including spinach, greens, kale, pac choi, and tatsoi, during early vegetative stage under increased salinity [86]. Moreover, fodder beet is known to tolerate up to 150 mM salt stress during the vegetative stage [16]. A sugar beet cultivar, O68, under salt stress, was also found to have better seedling growth than unstressed plants [85]. According to Naguib et al. (2021), salt tolerance strategies at seedling stage of sugar beet include partitioning of photosynthate to new developing leaves, equal distribution of Na^+^/K^+^ in leaves and roots, and raffinose accumulation in leaves [61]. However, changes in Na^+^ concentrations and Na^+^/K^+^ ratio are found at different developmental stages. In a recent report, Na^+^ concentrations increased in the roots of young sugar beets, but the reduction in Na^+^/K^+^ ratio was recorded at subsequent stages [50].

In addition to salt stress, low soil moisture contents at critical developmental stages such as field emergence and initial seedling establishment could be also detrimental for sugar beet growth [87]. Occurrence of drought stress at vegetative phase can dampen the root and shoot fresh weights and root diameter of sugar beet [8], suggesting that the yield reduction due to drought stress is reciprocated with plant developmental stage as well as severity of water limitation [88]. In a recent work, Skonieczek et al. (2018) identified three sugar beet cultivars, which can sustain low moisture at the emergence stage [89]. They found that the tolerant group conspicuously displayed faster and uniform seedling emergence compared to the sensitive group under low moisture, and it was suggested that the variation in seedling establishment between the tolerant and sensitive cultivars might be due to the differential responses under water deficit [89]. In red beet, highly significant reduction in forage yield was exhibited at maturity and root formation stages in response to water deficit [23].

As the beets are sensitive to stressful conditions during emergence and stand establishment, the beet varieties with better germination and stress tolerance under salt and drought should be developed for phytoremediation purposes [25]. For crop production in stressful conditions, identification of the tolerance levels of beet varieties, especially at early stages of seedling development, is required [25].

### 2.2. Osmotic Adjustment through Accumulation of Compatible Solutes

Plants counteract saline environments and water deficit through an internal osmotic adjustment attained by accumulation of compatible solutes, which enable them to cope with ion toxicity and maintain water uptake and cell turgor [90]. Compatible solutes or osmoprotectants are small, non-toxic molecules that protect cells against stress and function in reactive oxygen species (ROS) scavenging, maintaining membranes and protein structures [91]. Accordingly, in beets, increased levels of compatible solutes, including glycine betaine, proline, glucose, fructose [49,68], raffinose [61,72], and sucrose [71,77], during salt or water stress, maintain photosynthesis and stomatal conductance [70,92]. Under water stress, accumulation of osmoprotectants reduces the osmotic potential in sugar beet cells [40,71], facilitating water influx [18]. In addition, in sugar beet shoots and wild beet leaves, inorganic ions such as K^+^, Na^+,^ and Cl^−^ are involved in osmotic adjustment in response to salinity stress [59,68,70,90]. Roots can not absorp water efficiently due to the high osmotic pressure in saline conditions, resulting in physiological drought [21,93], and they must exclude almost all the toxic ions (97–98%) from the transpiration stream while absorbing water. Only 2–3% of NaCl helps plants osmotically adjust the Na^+^ and Cl^−^ in vacuoles [90]. However, low salt concentrations induce growth of sugar beet, red beet, and Swiss chard plants because of the role of Na^+^ in osmotic adjustment [35]. Importantly, the halophytic traits of beets are imparted to their ability of osmotic adjustment through accumulation of compatible solutes in cytoplasm and ions in the vacuoles of shoots in response to salinity or high osmotic pressure [59,70,84]. Lv et al. (2019) suggested that osmotic adjustment might be the most important trait for salt tolerance in beets [21]. In addition, many genes involved in compatible solute transport or biosynthesis were identified in beets, and their expression profiles were examined under salt and drought stress. Here, we describe the changes of compatible solute contents and gene expression profiles in response to salt and drought stress.

In plants, proline is one of the important amino acids for adjustment of osmotic potential under various environmental stresses, as well as for maintaining integrity of membranes and stabilizing the structure of proteins as a molecular chaperone [94]. In general, proline is accumulated in cytosol, and it is responsible for osmotic adjustment in cytoplasm [95]. The proline accumulation increased rapidly in different organs of sugar beet, fodder beet, and red beet under salt and drought stress [16,50,71,83]. For example, in a recent work, it was shown to increase remarkably in leaves but not in roots under high salt stress [55]. However, fodder beet showed increased proline content in both taproots and leaves under salt stress [16]. Moreover, the proline levels were significantly increased in salt-tolerant beet cultivars compared to sensitive ones under salt stress [51,67,70]. Like cultivated beets, wild beet can also accumulate proline in taproots under salinity [31]. Consistently, the expression of *δ-1-pyrroline-5-carboxylate synthase* (*P5CS*) gene encoding P5CS enzyme for proline synthesis was found to increase in *B. maritima* under high saline conditions [31]. By contrast, in *B. vulgaris,* they did not observe any increments in the gene transcription [31]. According to this study, the P5CS enzyme appears to be involved in salt tolerance of wild beet. Similar to gene expression profiles, although proline accumulation was observed in leaves under salt stress, P5CS protein did not increase in roots and leaves of sugar beet [53]. This suggests that sugar beet might have lost its ability to maintain proline contents under high salinity [31]. In addition to this, betaine/proline transporters (Bet/ProTs) were isolated from beets that function in betaine, proline, and choline transport [96]. The *BvBet*/*ProT1* gene was isolated from sugar beet, and its expression was slightly enhanced after salt treatment only in young leaves as compared to the plants under control conditions [97]. Similar to salt-treated beets, sugar beets under drought stress also accumulate higher amounts of proline in all plant organs, including storage roots [9,71]. Drought-tolerant sugar beet cultivars exhibited higher proline levels in leaves compared to the sensitive cultivars in response to water deficit [30]. However, it was demonstrated that the transcript levels of *BvBet*/*ProT1* and *BvBet*/*ProT2* genes were enhanced by drought stress only in *B. maritima* among drought-tolerant fodder beet, sugar beet, and wild beet genotypes. The drought-tolerant sugar beet genotypes did not even show up-regulation of these genes [18]. These findings suggest that wild beet plants are protected by proline and betaine against drought stress, but comprehensive studies need to be done on the roles of Bet/ProTs in different beet cultivars. In a very recent study, Ghaffari et al. (2021) found differences in proline concentrations of leaves and roots. The concentration was higher in roots than in leaves after water stress [11]. Interestingly, up-regulation of proline was observed not only under drought stress but also after rehydration [98]. In plants, exogenous proline improves root surface to handle water deficit and nutrient deficiency [99]. Consistently, recent reports ascertained that the exogenous application of proline can minimize drought-induced damages in sugar beet [9,11,54]. For instance, proline treatment brought about significant increase in root and sugar yield, the percentage of sucrose, chlorophyll content, and RWC but decreased ROS production, lipid peroxidation, and electrolyte leakage, protecting the plants against water stress [54]. Similarly, Ghaffari et al. (2019) also found that proline application led to a remarkable increase in internal proline content, as well as activities of antioxidant enzymes such as ascorbate peroxidase, catalase, and peroxidase, in response to drought stress [9]. As a result, proline is a beneficial parameter for stress evaluation in beet populations.

Soluble sugars (sucrose, fructose, and glucose), which take part in maintaining cellular osmoticum, may help to reduce leaf temperatures under salt and drought stress in beet genotypes [61,100]. Sugar beet is a rich source of sucrose and raffinose. As a non-reducing sugar, sucrose plays a key role in stress response, osmotic adjustment, the stabilization of membranes and proteins, and the prevention of protein denaturation [101]. In *B. vulgaris*, sugar transporters such as vacuolar sucrose importer (BvTST2.1) and sucrose transporter 1 (BvSUT1) were identified [102,103]. The BvTST2.1 was found to be involved in vacuolar sugar uptake in taproots of sugar beet [103], and BvSUT1 is required for sucrose loading into the phloem of sugar beet leaves [102]. Sugar transporters in different plants, including *Arabidopsis*, rice, barley, and soybean, are known to be involved in abiotic stress response [104,105,106]. However, we still do not know how sugar transporters in beets respond to salt or drought stress. Under environmental stress, sucrose content depends on the duration of stress; sugar beet genotype; and organ. For example, the content of sucrose in sugar beet leaves was increased after 3 h and 14 d of salt stress [107]. In another study, at 30 days after salinity, the salt-susceptible genotype (LKC-2006) enhanced sucrose content in roots compared to the control groups and reduced with increasing stress duration. In contrast to the LKC-2006, higher sucrose accumulation was observed in the roots of salt-tolerant sugar beet genotype (LKC-HB) after 60 days of salt stress [61]. Moreover, in the sugar beet cultivar (O68), the soluble sugars remarkably enhanced in both leaves and roots at 300 mM NaCl, but the elevation was higher in leaves than in roots [55]. However, two recent studies have shown reduced sucrose content in sugar beet roots after 1 day of salt stress and in LKC-HB leaves during salinity [50,61]. Liu et al. (2020) reported that the reduction of sucrose in roots might be due to the decomposition of sucrose into other soluble sugars and increments of tricarboxylic acid (TCA) cycle activity under stress [50]. Recently, sucrose synthases (SuSy), which take part in sucrose synthesis and decomposition in plants, have been found to be accumulated in sugar beet roots under salt stress, suggesting that osmotic regulation in roots may be related to the accumulation of these enzymes [53]. Furthermore, some studies have shown that augmentation of compatible solutes could also reduce sucrose accumulation in roots due to the assimilation of partitioning adjustments between storage carbohydrates and structural carbohydrates [68,77,108]. In contrast, it has recently been shown that enhancement of soluble sugars in sugar beet leaves is observed when plants are exogenously treated with proline [9]. In addition to sucrose, raffinose family oligosaccharides are also known as compatible solutes and have important roles in plant tolerance to several abiotic stresses including salt and drought, etc. [109]. Naguib et al. (2021) found elevated levels of raffinose in leaves of the salt-tolerant beet genotype (LKC-HB) under salinity stress [61]. Kito et al. (2018) isolated and characterized three sugar beet genes, *raffinose synthases* (*BvRS1* and *BvRS2*) and *galactinol synthase* (*BvGolS1*), which are all responsible for raffinose biosynthesis. The transcript levels of *BvRS1* gene in roots were marginally induced by salinity, whereas the *BvRS2* gene expression was upregulated after 3 days of salt stress in sugar beet leaves and roots [72]. In the study, raffinose accumulation was dramatically higher in roots than in leaves [72]. Similar to the raffinose levels, salt-tolerant genotype had significant increments in the transcript levels of *BvRS2* and *BvGolS1* genes in leaves and roots after salt stress [61]. As a result, the authors suggested the role of raffinose in increasing salt tolerance and shoot dry weight maintenance in sugar beet [61].

In addition to proline and soluble sugars, glycine betaine or betaine (GB) as a non-toxic osmolyte [110] and stabilizer of macromolecules [53] is significantly accumulated in different organs of sugar beet, red beet, and Swiss chard under salt and drought conditions [30,47,67,111]. Under high saline conditions, an increase in GB accumulation was reported in red beet/sugar beet leaves [47,67], sugar beet roots [50], and in all tissues of sugar beet [97]. Interestingly, even under non-stressed conditions, the sugar beet and Swiss chard were found to accumulate high amounts of GB [97]. Even the seeds of sugar beet accumulate GB, while *Arabidopsis* seeds do not, indicating the importance of GB on sugar beet seed vigor under salinity [84]. Substantially, higher levels of GB than proline in Swiss chard and sugar beet might contribute to the osmotic adjustment under different salt concentrations [33,111]. Under stress, drought- or salt-tolerant sugar beet cultivars showed higher amounts of GB in leaves [30,67], shoots, and taproots [42] compared to the susceptible cultivars. Two enzymes, ferredoxin-dependent choline monooxygenase (CMO) and betaine aldehyde dehydrogenase (BADH), found in chloroplasts, are responsible for GB biosynthesis [112]. CMO converts choline into betaine aldehyde [113], and then the second enzyme, BADH, generates GB [110]. Sugar beet genotypes with lower or higher salt/drought tolerance showed higher expression levels of *BvCMO* gene [18,67]. In recent studies, *BvCMO* gene expression in the guard cells and roots of sugar beet was significantly up-regulated under salt stress [50,64]. Zhang et al. (2008a) reported that tobacco plants expressing *CMO* gene from *B. vulgaris* (*BvCMO*) showed salt and drought tolerance by increasing GB levels. In addition, GB-accumulated tobacco plants exhibited elevated apparent quantum yield as well as net photosynthetic rate under salt stress [114]. The accumulation of high levels of GB in chloroplasts protects the thylakoid membranes and ultimately photosynthesis in response to water stress [18]. In accordance to the previous research data, transgenic sugar beet plants harboring the antisense *BvCMO* gene displayed higher salt stress sensitivity and reduction in GB synthesis activity from choline and BvCMO protein levels compared to the WT. However, no changes in GB contents between transgenic plants and WT were detected in old leaves [96]. Skorupa et al. (2019) reported that the expression of *BADH* gene increased in *B. maritima* plants acclimated to high salinity. However, in *B. vulgaris,* high salinity did not cause elevations in the transcription of this gene [31]. These results suggest that wild beets have a better performance in maintaining GB levels than sugar beet under salt stress. Opposite results were observed in sugar beet plants. For instance, the *BADH* gene expression was elevated 2–4 fold in both leaves and roots in sugar beet under 500 mM NaCl as compared to the control plants, indicating the strong salt stress responsive functions of *BvBADH* gene [115]. *BvBADH7* gene expression was also induced by salt stress in the salt-tolerant sugar beet genotype [67]. Similarly, a drought-tolerant sugar beet genotype displayed significantly enhanced transcription of *BvBADH1* gene under drought, but no changes in *BvBADH2* gene expression were observed. In contrast, the drought-sensitive one had increased *BvBADH2* gene expression in comparison to control and tolerant genotype [18]. These findings obtained from previous studies confirm that *CMO* and *BADH* genes are involved in salt and drought tolerance of sugar beet and wild beet plants.

Overall, the beet plants can adapt to salt or drought stress through accumulation of compatible solutes in different organs. The results summarized above indicate that the stress-tolerant cultivars increase compatible solute biosynthesis through regulating the expression of genes. Moreover, the identification of sugar transporters, and genes related to compatible solute biosynthesis, will help us to better understand stress tolerance strategies in beets and enhance sugar yield and abiotic stress tolerance for crop improvement programs.

### 2.3. Ion Transport

Cultivated beets accumulate salt ions in order to maintain ion homeostasis under salt stress. For instance, red beet is capable of accumulating higher levels of Na^+^ ions than other plants such as soybean, lettuce, rice, and bean [76]. Na^+^ and Cl^−^ ions were accumulated in the petioles and older leaves of sugar beet that may protect photosynthetic apparatus from ion toxicity through blocking the accumulation of salt ions in the young leaves [33]. The wild beet tends to shed its old leaves to prevent the internal accumulation of toxic Na^+^ and Cl^−^ ions in the younger and metabolically active parts to maintain K^+^ homeostasis [75]. Moreover, two contrasting sugar beet cultivars showed no differences in leaf Na^+^ amounts except for the distribution of Na^+^ among the fractions of leaves such as apoplasmic fluid, cell wall, and cell sap differed in response to salt stress [73]. During pot experiments, the levels of Na^+^ and K^+^ in sugar beet leaves and taproots increased under salt and drought stresses, respectively [70,77]. Relatively high NaCl levels, which induce growth in Swiss chard (*B. vulgaris* L. var. *cicla*), bring about accumulation of high Na^+^ contents in the leaves [116], but accumulation of other cations was restricted [117]. The fodder beet plants grown on chloride (Cl^−^)-containing soils of New Zealand showed higher root yield and sugar content with increased concentrations of NaCl. The fact that application of NaCl increased Cl^−^ content without altering Na^+^ or K^+^ concentrations in fodder beet organs helped to maintain ion homeostasis under salinity [118]. In salt-treated Swiss chard, significantly more Cl^−^ than Na^+^ ions accumulated, suggesting that lower Na^+^ in the cytoplasm might be due to Na^+^ efflux and Cl^−^ influx [111].

Various plant transporters and aquaporins (AQPs) take part in stress response and tolerance mechanisms. AQPs function in transport of water and small neutral molecules [119], and Na^+^ and K^+^ transporters are involved in maintaining the Na^+^/K^+^ ratio under stress conditions [120]. In sugar beet, the expression of some genes encoding proteins for ion channels such as *high affinity K^+^ transporter* (*HKT1*) or *potassium channel* (*KAT1*) was decreased under salinity [31], which may restrict the influx of toxic ions [120]. In order to overcome high saline conditions, the wild beet has evolved with various structural and physiological strategies to distribute salts and other solutes and enhance water content availability [62]. Plasma membrane intrinsic protein (PIP) channels mediate Na^+^ influx into roots [121]. In wild beet, the expression of *plasma membrane aquaporin* (*PIP*) genes *PIP1;1*, *PIP2;1*, and *PIP2;2* decreased after prolonged salt treatment, whereas short-term salt led to an increase in their transcription, implying the plasticity in the transcription of *AQP* genes in *B. maritima* [122]. Moreover, *B. vulgaris* showed a decrease only in the expression of *BvPIP2;2* gene, but the transcript levels of *BvPIP1;1* and *BvPIP2;1* remained unchanged under salinity stress [122]. The reduction in the transcript abundance in *B. maritima* and *B. vulgaris* might be related to the maintenance of water content during prolonged salt stress [122]. AQPs are known to express in leaf tissues and thus their expression patterns or activity may influence leaf water status under environmental stress. The relationship between AQPs and root hydraulic conductance has also been shown in previous studies. A decrease in *AQP* gene expression leads to the decline in root hydraulic conductivity, which is required for stomata closure, deceleration of transpiration rate and therefore adaptation to low water availability under salt stress [123]. In a recent study, PIP2 protein in red beets under saline conditions changed its unphosphorylated/phosphorylated abundance, which is also associated with the alterations in root hydraulic conductivity [52]. More research studies are needed to identify beet *AQP* genes and understand their contributions to salt and drought stress tolerance in wild beet varieties and cultivated beets. Moreover, the interactions between AQPs and other membrane proteins are still unknown in beets exposed to stress conditions. 

Sequestering of Na^+^ ions in the vacuole is one of the major strategies of Na^+^/H^+^ antiporters (NHXs) to maintain the Na^+^ homeostasis [124,125] and to dampen the toxicity of Na^+^ ions in cytoplasm leading to salinity tolerance of the plant [126]. The high activity of Na^+^/H^+^ exchange (NHX) at the tonoplasts provides salt tolerance in beets [127,128]. Na^+^/H^+^ antiport in beets was identified from tonoplast vesicles of storage tissues in red beet and sugar beet [125]. Accordingly, a NaCl-inducible vacuolar *NHX*, *B. vulgaris NHX1* (*BvNHX1*) gene was identified and characterized by Xia et al. (2002), which showed homology to *Arabidopsis NHX1* [129]. The transcript abundance of *BvNHX1* increased under salt stress. Moreover, the vacuolar Na^+^/H^+^ antiporter activity and BvNHX1 protein levels also enhanced after NaCl treatment, suggesting the salt tolerance role of BvNHX1 [129]. In a recent study, five putative Na^+^/H^+^ antiporter genes (*BvNHXs*) were identified in sugar beet, which were grouped into three classes, e.g., Vac-(*BvNHX1, −2* and *−3*), Endo-(*BvNHX4),* and PM class-NHX (*BvNHX5*/*BvSOS1*) [130]. Their transcript levels were markedly increased under high salt concentration in both roots and leaves. According to the prediction studies based on protein–protein interactions, only BvNHX5 interacts with calcineurin B-like protein (CBL) and CBL-interacting protein kinases (CIPK), suggesting that it might be the primary NHX under salt stress associated with CBL-CIPK pathway. Additionally, the presence of one abscisic acid (ABA) responsive element (ABRE) in *BvNHX5* shows the possible involvement of *BvNHX5* in ABA signaling pathway [130]. Previous findings on the regulation of vacuolar *NHX1* gene (*BvNHX1*) from salt-tolerant beets showed that a 337 bp promotor fragment is indispensable for the interaction of *BvNHX1* with its downstream interacting components. *MYB* class of transcription factors are the major interactors with the *cis*-acting elements within the 337 bp promotor fragment, which activates the expression of *BvNHX1* upon salt and water stress [131]. Transgenic studies were also performed to ascertain the functions of *NHX* genes in beets under stress. Transgenic sugar beet plants containing *Arabidopsis NHX1* (*AtNHX1*) gene showed an improved salt tolerance [132]. Furthermore, transgenic sugar beets harboring *Arabidopsis NHX3* (*AtNHX3*) gene also exhibited enhanced salinity resistance, and high soluble sugar content [133].

Plasma membrane H^+^-ATPases (PM H^+^-ATPases), which form an electrochemical gradient to regulate ion transport, play a substantial role in salt tolerance by H^+^ transport from cytosol to apoplast [134]. The hydrolytic and pumping activities of H^+^-ATPases remain unaltered with unaffected apoplastic pH under salt stress (150 mM NaCl), showing that sugar beet plants can adapt to salinity by regulating ion movement [134]. However, PM H^+^-ATPase activity decreased as a result of higher Na^+^/K^+^ ratio in the shoots of salt-tolerant sugar beet and salt-sensitive maize under salt, but sugar beet PM H^+^-ATPases were relatively more effective during low salinity (25 mM NaCl) compared to the maize [78]. Overall, the findings described above suggest that H^+^-ATPase activities in beets depend on the severity of salt stress. Moreover, some proteins such as vacuolar H^+^-ATPase (V-ATPase) and H^(+)^-transporting pyrophosphatase (PPase) were detected in the membranes of salt-grown beets [135]. Salt stress induced an increased expression of *V-ATPase* in sugar beet leaves under high salinity [124]. In previous studies, it has been shown that the reduction of PM H^+^-ATPase activity induces stomatal closure in different plants under water deficit [136], but the involvement of H^+^-ATPases, and transporters in drought stress tolerance is still unknown in beets.

### 2.4. Antioxidative System

Accumulation of reactive oxygen species (ROS), including superoxide (O_2_^**˙**¯^), hydrogen peroxide (H_2_O_2_), hydroxyl radicals (OH^•^), and singlet oxygen (^1^O_2_), is a well-known phenomenon in plants under unfavorable conditions. ROS molecules elicit oxidative stress and cause irreversible damage to nucleic acids, proteins, pigments, and lipids [137]. To alleviate ROS-induced damages, plants were equipped with several enzymatic and non-enzymatic antioxidants [138]. In plant cells, enzymatic antioxidants, which are responsible for scavenging of superoxide radicals and H_2_O_2_, include superoxide dismutase (SOD), catalase (CAT), peroxidase (POX), and ascorbate peroxidase (APX) [138,139]. Despite the existence of osmoprotective mechanisms in beets, prolonged drought and salt stress induce the accumulation of ROS such as O_2_^**˙**¯^ and H_2_O_2_ [50,54,73] and oxidative stress. In numerous studies, alterations in antioxidant enzyme activities and gene expression profiles under salt stress were shown in beets. For instance, under salt stress, sugar beet plants displayed higher activities of antioxidant enzymes such as SOD, CAT, POX, APX, and glutathione peroxidase (GPX), compared to the unstressed plants [33]. In another report, wild beet showed enhanced activities of antioxidant enzymes (SOD, POX, APX, CAT, and glutathione reductase (GR)), as compared to sugar beet, indicating that salt-tolerant *B. maritima* might have a better ROS scavenging capacity [4]. In a sugar beet cultivar (HI-0473), the activities of CAT and POX enzymes markedly increased under mild and severe salt concentrations [35]. Although SOD and POX activities increased in both leaves and roots of sugar beet, an increase in CAT activity was higher in leaves than in roots, and the activity remained high during severe salt stress [55]. However, the APX activity was decreased in response to severe salinity (150 and 250 mM NaCl), whereas lower salt concentrations (75 and 100 mM NaCl) enhanced the enzyme activity [35]. The differences in the activities of antioxidant enzymes, including CAT, APX, and POX, might enhance the levels of H_2_O_2_ and lower the stress tolerance in sugar beet plants. In the salt-tolerant sugar beet cultivars, the SOD, APX [51], and POX activities were elevated when compared with the sensitive cultivars, showing the contribution of antioxidant enzymes to the salt tolerance in stress-tolerant beets [51,67]. Consistently, under high salt concentrations, the transcript levels of *Cu-Zn-SOD*, *Mn-SOD*, *Fe-SOD3*, *alternative oxidase* (AOX), and *peroxiredoxins* (Prx) were increased, but ROS-generating NADPH oxidases were dramatically decreased in sugar beet [140]. In addition, Dunajska-Ordak et al. (2014) isolated a *peroxisomal ascorbate peroxidase*, *BvpAPX* gene, from *B. vulgaris* leaves [141]. The gene expression patterns obtained in the study are consistent with previous findings, which have shown elevated APX enzyme activity in stress-tolerant beet cultivars. In both *B. maritima* and *B. vulgaris*, a remarkable increase in the expression of *APX* was recorded after prolonged salt stress, but no changes were reported during short-term salt stress [141]. The enzyme activity of POX and its mRNA abundance in the roots and leaves of sugar beet were increased in response to salt stress [50,74]. Through ChIP assay, Yolcu et al. (2016) found that *POX* gene activation was associated to the enhanced levels of two histone H3 lysine acetylation types, H3K27 and H3K9, after salt stress in sugar beet and wild beet, respectively [74]. It is the first report indicating the relationship between chromatin modifications and salt stress response in sugar beet and wild beet. However, the effects of epigenetic modifications on the transcription of antioxidant-encoding genes in beet cultivars remain elusive. In beets, antioxidative system is one of the important stress coping strategies under abiotic stress, which was also established through several reports on beet transcriptomic analyses and transgenic lines. For instance, Li et al. (2020) generated transgenic *Arabidopsis* plants harboring *monodehydroascorbate reductase* (*MDHAR*) gene from sugar beet M14 line in order to investigate the contribution of *MDHAR* gene to salt stress tolerance [142]. The MDHAR is an antioxidative enzyme, which plays a key role in regulating ascorbate levels and thereby reducing the ROS accumulation [139]. Overexpression of the gene in *Arabidopsis* showed salt-tolerant phenotypes compared to the WT plants. The results supported that the *MDHAR* can be used as a promising candidate for improvement of stress-tolerance in plants [142]. In addition, through transcriptomics approach, sugar beet genes encoding late embryogenesis abundant (LEA) and POX proteins were shown to increase after salt stress [50]. LEA proteins are known to participate in stress tolerance, and osmoprotection, and they also act as antioxidants and chaperons in response to abiotic stress [143]. Moreover, they induce antioxidant encoding gene transcription and activities of antioxidant enzymes such as SOD, APX, and CAT, in rice [144]. Comprehensive studies are needed to better understanding the role of LEAs in stress tolerance of beet cultivars. In addition to antioxidant enzyme assays and gene expression analyses in beets, antioxidant enzyme-dependent growth stimulation under salinity was shown by Takagi and Yamada (2013) in Swiss chard [117]. A significant correlation was observed between dry matter production and the activity of CAT and APX enzymes under salt stress [117].

In the literature, drought stress-induced changes in antioxidant enzyme activities and their gene expression profiles were also detected in beets. The cultivar-specific variations were observed in antioxidant enzyme activities under water-deprived conditions. Recently, it was reported that the water deficit led to decrease in POX activity [8] but increase in SOD and CAT activity in sugar beet plants [8,54]. Contrastingly, in a previous work, CAT was found to be the most sensitive ROS scavenging enzyme among all the antioxidant enzymes in sugar beet under drought, as rehydration enhanced CAT enzyme activity [98]. In different sugar beet cultivars and Swiss chard, SOD, CAT, and POX enzyme activities remarkably declined under drought [30,145]. The authors reported that sugar beet cultivars with high lipid peroxidation showed lower antioxidant enzyme activities [30]. In gene expression analyses, the sugar beet genotype (DH0962) with highest drought tolerance exhibited a dramatic reduction in the expression of *BvSD1* gene encoding a Cu-Zn superoxide dismutase 1 [18]. By contrast, drought-tolerant beet cultivars had elevated activities of CAT, APX, and POX enzymes under drought [30]. Up-regulation of enzyme activities are stimulated by high oxidative stress, which is induced by H_2_O_2_ accumulation and lipid peroxidation [30]. However, in another study, the activities of APX, GPX, and CAT did not alter in sugar beet lines in response to water stress [20]. Interestingly, the expression levels of *peroxisomal ascorbate peroxidase 3* (*BvAP3*) gene increased in drought-sensitive sugar beet genotype during drought stress [18]. Compared to salt stress-related studies, less information is available on the contribution of antioxidants to drought stress response in beets, and therefore more research needs to be done in different beet cultivars. 

Collectively, all research data described above suggest that variations in antioxidant enzyme activities and transcript levels of antioxidant encoding genes depend on the beet variety used, stress conditions, and organ [30,35]. Alterations in the antioxidant enzyme activities, and gene expression profiles under salinity and drought stresses in beets, could be highly efficient strategies to surpass stress-induced damages while maintaining homeostasis [43,140].

### 2.5. Selection of Salt- and Drought-Tolerant Beets Based on Different Parameters

By compiling and classifying the morpho-physiological or genetic traits of existing beet cultivars under drought or salt stress conditions, the breeders can achieve robust strategies for generating new salt- and drought-tolerant cultivars [14,30,40,42]. Developing stress-tolerant beet cultivars might result in higher yield, higher productivity, and larger areas under cultivation [11]. In order to overcome the stress-induced yield loss in sugar beet, plant breeders should develop cultivars with high germination and establishment despite stress occurrence. Therefore, breeding process of stress-tolerant cultivars is time-consuming and costly [11].

Although different physiological traits have been used to select salt- and drought-tolerant beet cultivars, there are no established and universally accepted stress evaluation parameters except for root yield decrease [18,34,40]. Each research group selects the tolerant lines based on different parameters, including germination rate, root length [22,80], root yield, root/shoot ratio, white sugar yield, sugar content, and Na^+^ and K^+^ amounts in the roots [14,25,146]. Among them, root yield, white sugar yield, and sugar content are important parameters for beet production. Even though root yield decrease is a time-consuming and expensive selection criterion, it has been used to identify salt- and drought-tolerant sugar beet lines under stress conditions [18,146]. Previous studies have shown that drought stress causes a dramatic drop in sugar beet root yield and sugar yield [40,56,147]. The drought-tolerant lines were selected by the lowest decrease in root yield [18]. Mahmoud et al. (2018) reported that the drought-tolerant genotypes conferred higher dry matter accumulation, taproot growth, and sugar yields as compared to the sensitive ones [148]. The breeding programs mostly aim to enhance the content of sucrose [149], which is the principal form of white sugar in sugar beet roots, and its concentration determines the root quality. Sucrose yield, which is one of the most considerable traits for beets, defines the sucrose percentage in root weight minus loss during storage and processing [150]. The impacts of drought stress on sucrose yield vary according to physical properties of soil, climate conditions, stored soil moisture, and plant nutrition [151]. In addition, the white sugar yield depends on root yield and sucrose content [11,149]. Accordingly, the reduction of white sugar yield in sugar beet under water stress [147] might be related to the decline in root yield [11]. It was found that leaf area index (LAI) can be used as a selection criterion for drought-tolerant sugar beet cultivars that is tightly associated with root yield and sugar yield [40]. Because selection of beet varieties with highest leaf areas during early growth stage, it is considered as an effective way to augment sucrose yield [149]. In several studies, salt-tolerant and drought-tolerant beet genotypes can maintain higher leaf area compared to the susceptible genotypes under stress conditions [19,40,51]. However, in sugar beet genotypes under field conditions, water stress significantly decreased the LAI, due to the drought-induced leaf senescence [40]. Wild beets exhibit smaller leaf areas, which decrease the evaporation surface and ultimately increase water content and succulence under salt stress [62]. Similarly, a significant reduction in LAI was also observed in red beet cultivars [152] and drought-tolerant fodder beet in response to water stress [63]. Moreover, some other physiological attributes such as quantum efficiency of PSII (Φ PSII), osmotic potential [18], and absorption of photosynthetic active radiation (PAR) were also found to be associated with root yield and sugar yield [40]. 

Despite the different results on root morphology of beet cultivars [61,81], alterations in root length are important adaptive strategies for plant stress tolerance [11,25,153]. Sugar beet and fodder beet are deep-rooted crops [40,48,63], and under water deprivation, fodder beet is able to extract water from bottom soil layers [63]. However, sugar beet plants exposed to salt or drought stress displayed lower root length as compared to unstressed plants [54,153]. Among different sugar beet genotypes, salt-tolerant one (H30917) displayed longest root length as compared to other genotypes under salt stress [25]. Similarly, the root length has been found to increase gradually in the tolerant sugar beet genotype, LKC-HB, in saline conditions [61]. Moreover, saline and water-deficit conditions decrease root hydraulic conductivity, but *B. vulgaris* roots have capability of adjusting their root hydraulic conductivity to avoid water loss at the earlier stages of salt stress [52]. It was speculated that the tolerant cultivars tend to reduce the density of root tissue; thereby, they can improve axial hydraulic conductivity under water deficit [20]. To develop physiological selection criteria, Shaw et al. (2002) compared two sugar beet cultivars (i.e., 24367 and N6). Upon exposure to drought stress, the tolerant cultivar, 24367 produced more fibrous roots and displayed much reduction in shoot/root ratio and higher RWC levels compared to the sensitive cultivar, N6 [42]. In a very recent study, the shoot/root ratio decreased in sugar beet as the severity of salt stress increased [153]. Similarly, among salt/drought-tolerant wild beet varieties (CMP, OEI, and VMT), the CMP shows higher root/shoot ratio than the less tolerant ecotypes, which suggests that this wild beet ecotype can be useful for generating stress-tolerant beets [38].

So far, chlorophyll content, photosynthetic rate, and CO_2_ assimilation have been used as selection criteria in beets [18,19,34,40,53], as it is known that stress conditions damage the most stress-sensitive organelles (chloroplasts) and impact PSII activity and CO_2_ assimilation rate negatively [154]. In sugar beet, the decrease in chlorophyll levels caused by stress might be related to ROS production and osmotic stress, leading to pigment degradation, decrease in CO_2_ influx, and photosynthesis [40]. The chlorophyll content under stress in sugar beet may depend on the severity of salt stress. For example, it was declined by severe salt stress (250–300 mM NaCl) [55,73] but increased by mild salinity (75 and 100 mM NaCl), showing the improvement of photosynthesis in response to low salt concentrations [73]. Under salt and drought stress, reduction in transpiration results in an increment of temperature in sugar beet leaves [42,64]. Eventually, closure of stomota minimizes the water potential of leaves to maintain vital physiological functions such as photosynthesis and growth [48] at the cost of rise in leaf temperature [42]. Stomatal conductance is associated with the net photosynthesis rate, which indicates the accumulation of organic matter by photosynthesis [67]. Even small changes in stomatal conductance may cause large effects on water transport in plants [155]. The decrease in chlorophyll content, photosynthetic rate, and stomatal conductance was much lower in salt-tolerant beet cultivar (T710MU) than in sensitive one (S710) [67]. Similarly, Wang et al. (2019) also evaluated some morphological and physiological alterations to select salt-tolerant beet cultivars. Accordingly, they found that salt-tolerant sugar beet genotype displayed higher chlorophyll content and net photosynthetic rate than salt-sensitive genotype [51]. In a very recent study, it was reported that the drought-tolerant lines also displayed higher chlorophyll retention, and photosynthetic quantum yield [30]. Furthermore, the reduction of stomatal conductance and transpiration in seawater-treated wild beet (*B. maritima*) plants contributes to the maintenance of leaf turgor and, therefore, survival under salt stress [59]. The red beet (*B. vulgaris*) also decreased stomatal conductance in response to 200 mM NaCl [52]. In addition to succulence index, Φ PSII, and osmotic potential, Wisniewska et al. (2019) have used some other physiological traits such as petiole dry mass, leaf blade dry mass, blade area, and relative flavonoid content as selection criteria, which were accession-specific under drought conditions. Drought-tolerant sugar beet lines exhibited an increment in the contents of flavonoids [18], which could be an important selection parameter due to their antioxidant roles under abiotic stress.Assessing the variations of antioxidant enzyme activities among the beet varieties under stress conditions can be a good stress evaluation parameter [30,51,67]. Furthermore, estimating the concentrations of compatible solutes in plant cells can also be considered as a promising indicator to assess the salt and drought tolerance between beet genotypes [14,30,67,71]. In Figure 1, we demonstrate morpho-physiological, and biochemical parameters, used for selection of salt- and drought-tolerant beets.

To develop the stress-tolerant beet cultivars, the plant breeders must be able to develop universally accepted stress evaluation parameters. In addition to the morpho-physiological and biochemical traits, molecular selection criteria such as stress-inducible genes and DNA-based markers can also be explored to isolate stress resilient beets in the future.

### 2.6. An Overview of the Differences of Stress Responses in Cultivated Beets and Wild Beet

Wild beet (*B. maritima*) populations survive extreme conditions like salt marshes and seashore cliffs and thus show phenotypic variations [38]. Previous studies showed different wild beet varieties’ salt and drought tolerance ability through morpho-physiological and molecular analyses [4,17,18,38,41]. In addition to the physiological and biochemical changes of wild beet under salt and drought stress described in previous subsections, we briefly outline the differences of salt and drought stress response in cultivated beets and wild beet and summarize the evolutionary studies in beet populations.

Salt and drought stress responses in plants are similar. For instance, the early effect of salt stress (osmotic stress) is entirely identical to drought stress. Moreover, the existence of salt ions and water deficit both cause low water potential. Both stress factors bring about the limitation of water uptake; stomatal closure; reduction of growth; and ROS production. However, the impacts of salt and drought stresses vary according to the beet genotype [38]. Wild beet displays various salt-tolerance strategies, including high succulence index, higher volume of the palisade and spongy parenchyma cells, smaller leaf area, more number of leaves, osmotic adjustment, and higher antioxidant enzyme activities compared to cultivated beets [4,31,59,62,65,66]. Salt and water accumulation in wild beet results in succulence and alterations of leaf structure, such as increments of the palisade and spongy parenchyma cell volume [69]. In *B. maritima*, smaller leaf areas, which constitute a salt stress coping mechanism in halophytes [156], lead to changes in carbon assimilation rather than a decline in photosynthetic rate under salt stress [62]. In contrast to wild beet, plant breeders mostly select cultivated beet genotypes with the highest leaf areas to get higher sucrose yield in the selected genotype [149]. The adverse impact of salt stress on beet growth is due to the Na^+^ and Cl^−^ accumulation [70]. However, the wild beet can adapt to low water potential and high levels of Na^+^ and Cl^−^ [66]. Sugar beet and wild beet both maintain leaf turgor by reducing stomatal conductance and transpiration, accumulating excessive amounts of Na^+^ and Cl^−^ ions in leaves, and accumulating compatible solutes such as proline and sucrose under salt stress [59,66]. Wild beet is adapted to different environments by regulating leaf temperature to maintain the leaf water status [38]. Unlike wild beet, leaf temperature in the salt-sensitive sugar beet genotype (LKC-2006) was higher than the tolerant genotype, LKC-HB, under salt stress [61]. Pinheiro et al. (2018) compared the salt stress responses between a sugar beet var. Isella and three wild beet ecotypes (CMP, OEI, and VMT) grown in different locations such as salt marsh, coastland, and dry inland, respectively [17]. The ecotype specific variations were observed among the parameters viz. fresh biomass; total seedling length; and hypocotyl, root, and cotyledon lengths. For instance, salt stress declined fresh biomass of seedlings, hypocotyl, root, and cotyledon lengths in all the wild beet ecotypes and sugar beet. The decrease in biomass due to salt stress was lower in CMP and VMT than in sugar beet and OEI. Among the beets, sugar beet was the only one that showed an enhanced shoot/root ratio during salt stress. In contrast to sugar beets, the seeds of wild beet ecotypes germinate under high saline conditions [17]. Moreover, at the seedling establishment stage of development, wild beets can grow under salt stress with low photosynthesis capacity in cotyledons. Likewise, Rozema et al. (2015) also compared the salt tolerance of sugar beet cultivars and wild beet. They observed a higher relative growth rate (RGR) and better salt tolerance in wild beet than in cultivated beets at the elevated NaCl concentrations [41]. In addition, domestication is an important evolutionary process in beets, leading to formation of new domesticated plants from wild species [157]. To examine the domestication process from wild beet to modern cultivated beets, differentially expressed genes (DEGs) were analyzed in *B. maritima* and *B. vulgaris* in several studies [31,67,158]. Accordingly, a previous transcriptomic study in wild beet identified various DEGs related to membrane transport, osmotic adjustment, molecular chaperoning, redox metabolism, and protein synthesis under salt stress [158]. In addition to this, based on RNA-seq analysis, Skopura et al. (2019) described that photosynthesis inhibition, wax and cuticle deposition, and leaf and cell size reduction may have been acquired to combat salinity stress in sugar beet during domestication. The DEGs, which were only expressed in sugar beet, might implicate that these traits were inherited from wild beet [31]. Although wild beet shows higher salt tolerance than sugar beets, salt tolerance traits in sugar beet have not been negatively influenced by domestication [31,41]. In a recent work, the transcriptomic analyses identified the DEGs, which function in carbon metabolism and amino acid biosynthesis in sugar beet roots under salt stress [50]. This suggested that sugar beet displays salt tolerance by regulating carbon and nitrogen metabolism, rapidly activating the sugar metabolism under salt stress [50]. Geng et al. (2019) compared two contrasting sugar beet genotypes (T710MU and S710) in salt conditions [67]. Their analysis identified several single nucleotide polymorphisms (SNPs) found only in the salt-tolerant cultivar, T710MU, compared to the sensitive one, S710. These SNPs might contribute to the salt tolerance in the T710MU cultivar [67]. Hence, transcriptome data suggest that different beet cultivars stimulate different processes in response to salt stress. Further experimental evidence is needed to address which stress-tolerance traits were lost or modified during domestication and the molecular mechanisms behind how modern beet cultivars become less tolerant to stress than their wild progenitor [31].

Compared to salt stress, less information is available on the drought stress response mechanisms in wild beets. In a previous study, among wild beet ecotypes (CMP, OEI, and VMT) and a sugar beet var. Isella, the VMT showed better tolerance to salt and drought than other wild beets and sugar beet [38]. Drought stress affected wild beet performance more than salt stress, and wild beet ecotypes showed distinct responses to drought and salt stress. For example, VMT displayed higher root growth under drought stress, but the highest shoot/root ratio in response to salinity stress [38]. Drought stress generally impinges shoot growth more than root growth due to the osmotic adjustment ability of roots [159]. Similarly, in sugar beet, leaf growth is more vulnerable to drought stress than root growth [56]. The difference between root and leaf growth increases the storage root/shoot dry matter ratio in sugar beet plants [56]. By contrast, Hoffman et al. (2010) found the decline in storage root/leaf dry matter ratio in sugar beet under drought. The storage root dry mass decreases in response to drought stress. In a recent report, the lowest decrease was seen in *B. maritima* and fodder beet, but the maximum decrease was detected in less tolerant sugar beet lines [18]. In addition, a comparative work using sugar beet, wild beet, and fodder beet accessions showed that the fodder beets exhibit the best drought tolerance [18]. Still, *B. maritima* showed superior performance than fodder beets and sugar beets in specific leaf weight under drought conditions. As a protective mechanism, the increased specific leaf weight and succulence index in wild beet and fodder beets under drought are associated with increased leaf thickness, protecting leaves from heat [18,40]. Drought stress is associated with irradiation and heat, which cause the imbalance between ROS production and antioxidative defense, and consequently oxidative stress [12]. In sugar beet plants, the chlorophyll content is reduced due to ROS accumulation and osmotic stress under drought [40]. However, in a comparative study, chlorophyll content decreased most in less tolerant sugar beet lines than wild beet and fodder beets [18]. These results suggest that the physiological parameters such as shoot and root growth, and chlorophyll content, depend on the beet genotype and stress conditions. To increase our understanding of the stress-tolerance mechanisms in beets, comparative studies between cultivated beets and wild beets are needed to be performed.

Genetic markers such as microsatellites, AFLP, and RFLP markers have been used to examine the evolutionary dynamics of genetic variation in *B. vulgaris* and *B. maritima* [27,38,160]. Because sugar beet was obtained from a single population, it has a minimal genetic variation to fight environmental stress [27,160]. The domestication and breeding processes are also considered to cause low genetic variation in cultivated beets. Still, there are great variations in economic traits, including yield, biomass, and stress tolerance in sugar beet, as these traits are regulated by polygenes and environmental changes [18]. Even though wild beet (*B. maritima*) is the ancestor of all beets, during domestication process, modern sugar beet cultivars are believed to have originated from the crossing of fodder beet with chard. Naturalized introgressions of wild beet with cultivated beets generate ruderal beets with high genetic diversity, and they can be used to improve the beet genotypes [27]. However, gene flow from cultivated beets to their wild ancestor through hybridization process can decrease the genetic diversity of wild beets [161], which could potentially have stress tolerance traits [38]. Allele diversity and heterozygosity (genetic diversity) are higher in wild beet than in sugar beet cultivars [18,160] due to the existence of selective pressures like salt and drought in their habitats [27]. Interestingly, strong genetic divergence was found between wild beet and other relatives [160]. However, Saccomani et al. (2009) reported that Italian ruderal beets clustered more closely with sugar beet than wild beet [27]. By contrast, wild beet ecotypes (CMP, OEI, and VMT) are closer to each other than sugar beet [38]. AFLP analysis and morpho-physiological changes indicated that the wild beet and sugar beet accessions are grouped into distinct clusters [27,38]. Through the RAPD technique, sugar beet, wild beet, and fodder beet accessions were also screened for genetic diversity. The sugar beet genotype with the highest drought tolerance was found to have the maximum genetic similarity to wild beet and fodder beets [18]. The results summarized above suggest that stress-tolerant sugar beet populations appear to be closely related to wild beet. To broaden our understanding of beet evolution, morpho-physiological traits and molecular markers are needed to examine in different beet populations. Furthermore, the evolutionary origins of the wild beets and cultivated beet varieties from different locations should be investigated.

### 2.7. Molecular Mechanisms Mediating Salt or Drought Stress Response

Over the past few years, several regulatory mechanisms in plants pertaining to their responses to various abiotic stresses were uncovered while utilizing the advances in molecular and genomic approaches [162]. Putnik-Delić et al. (2017) demonstrated the differential expression of candidate genes involved in salt and osmotic stress response under drought and suggested that these genes can be used for development of DNA-based markers for sugar beet breeding [163]. Additionally, there are only two research studies describing the effects of epigenetic modifications such as histone acetylation and DNA methylation on gene expression in *B. vulgaris* and *B. maritima* under salt stress [74,164]. Due to insufficient information on the role of epigenetic mechanisms in abiotic stress response of beets, we have not discussed this molecular mechanism in the present review. In previous sections, we already mentioned the beet genes known to be involved in compatible solute biosynthesis, antioxidative defence system, and ion transport. Here, we have described noncoding RNAs and some of the beet genes related to salt or drought stress response/tolerance.

#### 2.7.1. Noncoding RNAs in Salt Stress Response of Beets

MicroRNAs (miRNAs) are endogenous small noncoding RNAs, ranging from ~19 to 24 nucleotides [165], which are shown to play important roles in growth, development, and various stress responses in plants [166,167,168]. The miRNAs regulate the expression of their target genes at post-transcriptional level via degradation, and at the post-translational level by inhibition [165,169]. Although the miRNAs are known to be activated in response to stress in plants, similar data from beets have been less abundantly reported [55,170]. Through bioinformatics approach, beet genome was found to encode 13 mature miRNAs, and their targets encode transcription factors, signal transduction components, and factors related to stress response [171]. However, the target genes of some of the sugar beet miRNAs, including *Bvu-miR4* and *Bvu-miR9~12*, have not been predicted so far [171]. Recently, Cui et al. (2018) found down-regulation of *miR160* and the corresponding up-regulation of its targets *Auxin Response Factor 17* and *18* (*ARF17* and *ARF18*) in beet varieties during salt stress. Moreover, the same study found expression of NAC transcription factors *NAC21*, *NAC22*, and *NAC100* as targets of *miR164* under salt stress exclusively in the salt-tolerant cultivar of sugar beet [170]. The expression of *miRNA160* and *miRNA164* varied significantly according to beet variety, stress duration, growth stage, and organ. Their results suggested that plants can adapt to high salinity conditions by inhibiting *miR160* and promoting the rapid release of *ARF17* and *ARF18* [170]. In a very recent study, the roles of non-coding RNAs (ncRNAs), including long non-coding RNAs (IncRNAs), miRNAs, and circular RNAs (circRNAs), were elucidated during salt stress in sugar beet cultivar, O68 [55]. In this report, whole-transcriptome sequencing of leaves and roots of sugar beet under salt stress to construct a competitive endogenous RNA (ceRNA) regulatory network demonstrated that the number of salt-responsive genes, including coding and ncRNAs, are higher in roots than in leaves [55]. In sugar beet, the ceRNAs are involved in numerous processes such as copper redistribution, plasma membrane permeability, glycometabolism, and energy metabolism. In summary, sugar beet roots get energy by increasing glycometabolism and fatty acid metabolism, but the leaves ensure photosynthesis to obtain the energy required to fight stress [55].

These reports summarized above have increased our understanding of the roles of ncRNAs in beets, and they might help scientists to improve tolerant beets with higher sugar and root yields. In the future, in order to understand the details of molecular mechanisms underlying the stress tolerance of beets, we need to examine the functions of the ncRNAs under salt and drought stress.

#### 2.7.2. Beet Genes Known for Their Involvement in Response to Salt and Drought Stresses

In beet genome, so far, only a few genes have been characterized and reported for their salt- or drought-stress-responsive roles compared to the model plant species. In addition to the beet genes described in previous sections, here, we briefed some of the genes reported for their involvement in salt and drought stress response. In Table 2, we list the stress-responsive genes in beets.

##### Basic/Helix–Loop–Helix 93 (BvbHLH93)

It was suggested that *B. vulgaris bHLH93* (*BvbHLH93*) gene encoding the basic/helix-loop-helix (bHLH) transcription factor is involved in salt response in sugar beet plants. Overexpression of this gene in *Arabidopsis* increased salt tolerance by lower Na^+^ and lipid peroxidation levels, higher activities of antioxidant enzymes, and lower transcript levels of respiratory burst oxidase homolog genes, *RbohD* and *RbohF* [46].

##### Sucrose Non-Fermenting-1-Related Protein Kinase 2 (SnRK2)

The sucrose non-fermenting-1-related protein kinase 2 (SnRK2s) is a protein belonging to Ser/Thr kinase family, found to be involved in growth, development, and abiotic stress response in plants [179]. In a recent study, *SnRK2* homologs were identified in sugar beet genome using bioinformatics approach and *BvSnRK2* transcript levels were augmented during salinity stress, showing their potential roles in salt stress response [172].

##### Cystatin

Plant cystatins encoding cysteine protease inhibitors were shown to be involved in abiotic stress tolerance [180]. Wang et al. (2012) isolated and characterized cystatin gene (*BvM14-cystatin*) from sugar beet M14 line for the first time. In their study, salt stress enhanced the expression of *BvM14-cystatin* gene in seedlings. Furthermore, *Arabidopsis* plants overexpressing *BvM14-cystatin* exhibited higher survival rates and less damage in primary root growth and consequently higher salt tolerance than WT [173].

##### S-Adenosylmethionine Decarboxylase (SAMDC)

The polyamines are low-molecular-weight molecules, known for their involvement in diverse processes, including development and stress response in plants [45,107,181]. The *S-adenosylmethionine decarboxylase* (*SAMDC*) gene encodes a key and rate-limiting enzyme for the biosynthesis of polyamines (spermine and spermidine) and was found to increase in the roots and leaves of sugar beet M14 line under salinity stress. *Arabidopsis* transgenic plants expressing the *SAMDC* gene displayed significantly high salt tolerance through increasing antioxidant enzyme activities and lower ROS production [45].

##### S-Adenosylmethionine Synthetase (SAMS)

S-adenosyl-L-methionine (SAM) synthetase, which is a precursor for polyamine synthesis [182], plays an important role in regulating metabolism, development, and stress response. Overexpression of *SAMS* gene from M14 line (*BvM14-SAMS2*) caused increased salt and oxidative stress (H_2_O_2_) tolerance in *Arabidopsis* by strengthening the antioxidative system and polyamine metabolism [175].

##### Glyoxalase I

Glyoxalase I enzyme is responsible for detoxification of methylglyoxal (MG), a cytotoxic by-product [183]. In sugar beet M14 line, the expression of *BvM14-glyoxalase I* gene was induced in response to different stresses, including salt, mannitol, and oxidative. Moreover, overexpression of *BvM14-glyoxalase I* in tobacco ameliorated tolerance to multiple stresses, including salt, mannitol, and H_2_O_2_ [176].

##### BETA1

In a previous work, *BETA1* gene, which is a homolog of *Arabidopsis SAH7* gene, was discovered by screening a cDNA library of *B. maritima* [178]. The *BETA1* gene expression was induced by salt stress in leaves and roots of wild beet. The function of this gene is unknown, but Uysal et al. (2017) showed that it might participate in salt tolerance in wild beet plants.

##### Serine O-Acetyltransferase (BvSAT)

Mulet et al. (2004) aimed to verify and isolate osmotic stress-responsive genes in sugar beet by randomly overexpressing them in osmotic stress-sensitive yeast strain [174]. From a cDNA library of the sugar beet, the *Serine O-acetyltransferase* (*SAT*) gene, which is important for the biosynthesis of cysteine in an alternative pathway, was found to have osmotic stress responsive function in sugar beet [174]. The *BvSAT* expression enhanced the production of low molecular weight containing sulphydryl molecules, which eventually rendered stress resistance in yeast [174].

##### Non-Symbiotic Hemoglobin (BvHb2)

Expression of *non-symbiotic hemoglobin* (*hb*) genes in plants are used as a strategy to cope with oxidative stress through enhancing antioxidant enzyme activities [184]. Non-symbiotic hemoglobin proteins in roots and leaves of sugar beet are involved in response to salt stress [53]. The *BvHb2* is a class 2 non-symbiotic hemoglobin gene, which is mainly expressed in leaf tissue and found to be responsive to light and osmotic stress in sugar beet [44]. Overexpression of *BvHB2* gene in *Arabidopsis* and tomato resulted in enhanced drought stress tolerance [44].

##### Heat Shock Factor (BvHSF)

Heat shock factors (HSFs) are a part of the large network of transcription factors whose expression is crucial for plant responses to various abiotic stress conditions [185]. The expression of *B. vulgaris heat shock factor* (*BvHSF*) gene was elevated under PEG-induced water stress, suggesting the involvement of *BvHSF* gene in drought stress response [177].

## 3. Conclusions and Future Prospects

Cultivated beets are economically important crops grown for the production of sugar; bioethanol; animal feed; and use in health, pharmaceutical, and food industries. However, the cultivation of beets is limited by adverse environmental constraints such as high salinity and drought. Fortunately, by virtue of its stress-tolerant traits, wild beet (*B. maritima* L.) can easily overcome stress conditions compared to other beet cultivars. Stress-tolerant beet germplasms such as M14 can also serve as a beneficial resource for enhancing food, yield, and bioenergy production in several different crops [7]. To gain a better idea of stress acclimation strategies in wild beets, more comparative studies need to be performed between the wild and modern genotypes. However, due to insufficient knowledge on stress tolerance mechanisms, and poor germplasm screening strategies, we were unable to adequately meet the goals of generating stress-resistant beet genotypes with higher yield. Still, we have learned many lessons from the morpho-physiological, biochemical, and molecular response mechanisms in stress-tolerant crops including beets. However, the details of physiological and molecular response mechanisms in beets are still unknown. Furthermore, there is little information about the interplay between metabolomic and transcriptomic responses to abiotic stresses in beets [50].

Genomics along with bioinformatics approaches to understanding the phenotypic diversity of beet cultivars under environmental stresses would be important for developing suitable breeding approaches and ultimately stress-tolerant beets. We believe that more knowledge on salt/drought tolerance strategies and stress-inducible genes in wild beet ecotypes and cultivated beets will enable plant scientists to improve selection parameters for generating tolerant beet cultivars in saline and dry soils. Moreover, the SNPs in the stress-tolerant alleles of beets should be detected to understand the relationships between stress tolerance and SNPs [67]. We need to perform comparative studies among the alleles from the tolerant and susceptible beet genotypes [186]. The application of modern biotechnological advancements like genome-wide association studies (GWAS), whole-genome surveys, and gene target surveys will open new opportunities and ease the prediction of causative elements at a single nucleotide level resolution [186]. Likewise, the application of forward genetic approaches can also uncover beneficial traits in stress-tolerant wild crop progenitors [186]. Additionally, employing the reverse genetic strategies and characterizing the T-DNA loss or gain of function mutants, and applying the revolutionary CRISPR/Cas mediated gene editing, is another alluring platform to achieve climate-resilient beet cultivars and generate modifications in sucrose transporter genes to enhance sugar yield [103].

## Figures and Tables

**Figure 1 plants-10-01843-f001:**
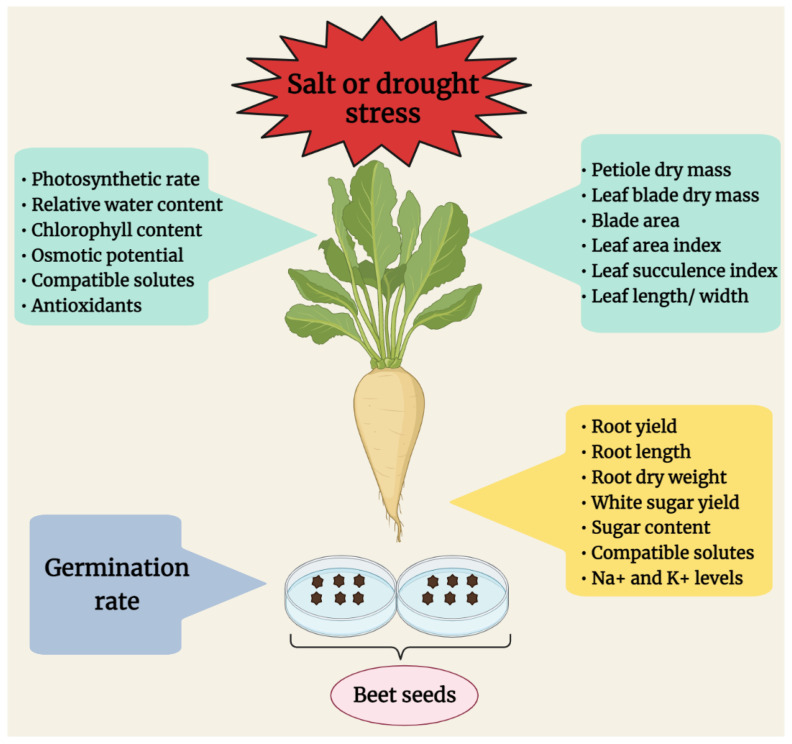
A brief summary of morpho-physiological and biochemical traits used to select salt-tolerant and drought-tolerant beets under high salinity and drought stresses. This figure was created by author in BioRender.com (12 July 2021).

**Table 1 plants-10-01843-t001:** A brief summary of the effects of salt and drought stress on cultivated beets (*B. vulgaris* L.) and wild beet (*B. maritima* L.) at morpho-physiological and biochemical levels. *Beta vulgaris* (cultivated beets) includes sugar beet, fodder beet, red beet, and chard.

Salinity Stress		Drought Stress		References
*Beta vulgaris* L.	*Beta maritima* L.	*Beta vulgaris* L.	*Beta maritima* L.	
A dramatic decline in germination and seedling growth	Capability of germination and seedling growth	A dramatic decline in germination and seedling growth	Capability of germination and seedling growth	[17,22,25,30,38]
Decline in the root weight and root length	Higher root/shoot ratio	Decline in the root weight and root length	Higher root growth	[8,34,38,61]
Low water content and small leaf area	Higher water content availability and smaller leaf area	Low water content and a dramatic decline in leaf area	Decrease in soil relative water content	[31,38,40,62,63]
High leaf temperature due to the reduction of transpiration	Low leaf temperature	High leaf temperature due to the reduction of transpiration	Low leaf temperature	[18,38,42,61,64]
-	Increments in specific leaf weight	Increments in specific leaf weight	Higher increments in specific leaf weight	[18,40,59,65,66]
Decrease in chlorophyll content, photosynthetic rate, and stomatal conductance	Decrease in photosynthetic rate, and stomatal conductance	A dramatic decrease in chlorophyll content, photosynthetic rate, and stomatal conductance	Decrease in photosynthetic rate, and stomatal conductance	[30,31,40,52,59,67]
High leaf succulence	Higher leaf succulence and higher volume of the palisade and spongy parenchyma cells	High leaf succulence in tolerant genotypes	Higher leaf succulence and higher volume of the palisade and spongy parenchyma cells	[62,65,68,69]
Accumulation of compatible solutes	Higher osmotic adjustment ability by compatible solutes	Accumulation of compatible solutes	Higher osmotic adjustment ability by compatible solutes	[18,31,70,71,72]
ROS accumulation and oxidative stress	Lower ROS accumulation and oxidative stress	Imbalance between ROS accumulation and antioxidants	-	[4,18,30,33,73,74]
Increase or decrease in the activities of antioxidant enzymes	Increased activities of antioxidant enzymes	Increase or decrease in the activities of antioxidant enzymes	Increased activities of antioxidant enzymes	[4,54,74]
Differences in the distribution of Na^+^ among leaf fractions	Preventing the internal accumulation of Na^+^ and Cl^−^ ions in young organs	Accumulation of Na^+^, K^+^ and Cl^−^ ions	-	[70,73,75,76,77]
Decline in plasma membrane (PM) H^+^-ATPase activity in the tolerant genotype	-	-	-	[78]

**Table 2 plants-10-01843-t002:** List of beet genes involved in salt- and drought-stress response.

Gene Symbol	Gene Product/Full Name	References
*Bet*/*ProT1*	Betaine/Proline transporter1	[18,96,97]
*Bet*/*ProT2*	Betaine/Proline transporter2	
*BADH*	Betaine aldehyde dehydrogenase	[18,31,61,64,67,72,96,114,115]
*CMO*	Choline monooxygenase	
*P5CS*	δ-1-pyrroline-5-carboxylate synthase	
*BvRS1/2*	Raffinose synthase 1	
*BvRS2*	Raffinose synthase 2	
*BvGolS1*	Galactinol synthase 1	
*Cu-Zn-SOD*	Copper-zinc superoxide dismutase	[18,50,74,140,141,142]
*Mn-SOD*	Manganese superoxide dismutase	
*Fe-SOD3*	Iron superoxide dismutase	
*POX*	Peroxidase	
*APX*	Ascorbate peroxidase	
*MDHAR*	Monodehydroascorbate reductase	
*AOX*	Alternative oxidase	
*Prx*	Peroxiredoxins	
*LEA*	Late embryogenesis abundant	
*HKT1*	High affinity K^+^ transporter	[31,122,124,129,130,131,132]
*KAT1*	Potassium channel	
*NHXs*	Na^+^/H^+^ antiporters	
*SOS1*	Salt-overly-sensitive1	
*PIPs*	Plasma membrane aquaporins	
*V-ATPase*	Vacuolar H^+^-ATPase	
*SnRK2*	The sucrose non-fermenting-1-related protein kinase 2	[172]
*Cystatin*	Cysteine protease inhibitor	[173]
*SAT*	Serine O-acetyltransferase	[174]
*SAMDC*	S-adenosylmethionine decarboxylase	[45,175]
*SAMS*	S-adenosylmethionine synthetase	
*bHLH93*	Basic/helix-loop-helix93	[46]
*Glyoxalase I*	Methylglyoxal detoxification	[176]
*Hb2*	Class 2 non-symbiotic hemoglobin	[44]
*HSF*	Heat shock factor	[177]
*BETA1*	-	[178]

## Data Availability

Not applicable.

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
