# Peer review of "Salt and Drought Stress Responses in Cultivated Beets (Beta vulgaris L.) and Wild Beet (Beta maritima L.)"

_plants, 2021, doi:10.3390/plants10091843_

Round 1

Reviewer 1 Report

The references  cited by authors are comprehensive and new.

But, need add some own views to comment  the results with contradictory as far as possible, and  reduce the stacking of references.

Please highlight the differences of  salinity and drought stress response  in cultivated beets and wild beet, and show the possible evolutionary path.

Please establish the relevancy among different indexes (including morphological,physiological,biochemical,and molecular indexes) as far as possible.

Reviewer 2 Report

The manuscript titled “  Salinity and drought stress response and tolerance mechanisms in cultivated beets (Beta vulgaris L.) and their ancestor, wild beet (Beta maritima L.)” is a comprehensive review of the genetic basis of salt stress tolerance in Beta spp. The review is well written, informative and the body of literature is very large. Just one minor change, L 203 change “immino” with “amino”

Author Response

Response to Reviewer 2 Comments

Comments:

The manuscript titled “ Salinity and drought stress response and tolerance mechanisms in cultivated beets (Beta vulgaris L.) and their ancestor, wild beet (Beta maritima L.)” is a comprehensive review of the genetic basis of salt stress tolerance in Beta spp. The review is well written, informative and the body of literature is very large. Just one minor change, L 203 change “immino” with “amino”

 Point 1: Just one minor change, L 203 change “imino” with “amino”

 Response 1: As per the suggestion, “imino” has been changed to “amino”.

Reviewer 3 Report

The paper is a good review regarding the effects of salinity and drought stress on beets, especially on sugar beet, fodder beet, red beet, Swiss chard (B. vulgaris L.) and wild beet (B. maritima L.).

The manuscript is based on huge number of papers published in important journals.

I have few comments:

The title contains lot of information which are do difficult to read. The authors should rephrase it in order to simplify it.

Keywords: I recommend avoiding the words from title.

Because the paper contains numerous aspects, I recommend to include two tables in which to synthetized the important effects of salinity, respectively of drought, on each beet cultivar.

Round 2

Reviewer 1 Report

The manuscript has been sufficiently improved to warrant publication in Plants